# A System for Standardizing and Combining U.S. Environmental Protection Agency Emissions and Waste Inventory Data

Ben Young [1], Wesley W. Ingwersen [2,*], Matthew Bergmann [2,3], Jose D. Hernandez-Betancur [4], Tapajyoti Ghosh [1], Eric Bell [1] and Sarah Cashman [1]

[1]  Eastern Research Group, Inc., Lexington, MA 02421, USA; ben.young@erg.com (B.Y.); tapajyoti.ghosh@nrel.gov (T.G.); eric.bell@erg.com (E.B.); sarah.cashman@erg.com (S.C.)
[2]  U.S. Environmental Protection Agency, Office of Research and Development, Cincinnati, OH 45268, USA; matthewbergmann@posteo.net
[3]  General Dynamics Information Technology, Inc., Falls Church, VA 22042, USA
[4]  Oak Ridge Institute for Science and Education, U.S. Environmental Protection Agency, Office of Research and Development, Cincinnati, OH 45268, USA; jodhernandezbemj@gmail.com
[*]  Correspondence: ingwersen.wesley@epa.gov

**Abstract:** The U.S. Environmental Protection Agency (USEPA) provides databases that agglomerate data provided by companies or states reporting emissions, releases, wastes generated, and other activities to meet statutory requirements. These databases, often referred to as inventories, can be used for a wide variety of environmental reporting and modeling purposes to characterize conditions in the United States. Yet, users are often challenged to find, retrieve, and interpret these data due to the unique schemes employed for data management, which could result in erroneous estimations or double-counting of emissions. To address these challenges, a system called Standardized Emission and Waste Inventories (StEWI) has been created. The system consists of four python modules that provide rapid access to USEPA inventory data in standard formats and permit filtering and combination of these inventory data. When accessed through StEWI, reported emissions of carbon dioxide to air and ammonia to water are reduced approximately two- and four-fold, respectively, to avoid duplicate reporting. StEWI will greatly facilitate the use of USEPA inventory data in chemical release and exposure modeling and life cycle assessment tools, among other things. To date, StEWI has been used to build the recent USEEIO model and the baseline electricity life cycle inventory database for the Federal LCA Commons.

**Keywords:** National Emissions Inventory; Toxics Release Inventory; substance registry service; facility registry service; eGRID; RCRAInfo; biennial hazardous waste report; discharge monitoring report; EPA data; greenhouse gas reporting program; python; tool ecosystem





## 1. Introduction

The U.S Environmental Protection Agency (USEPA) administers national programs for collection, verification, and distribution of information on individual facility generation of waste and releases as well as areal (e.g., U.S.) and sector-based emissions. Together, these programs compile data on releases of various types of pollutants as well as generation and management of hazardous and toxic wastes. In general, these programs publish public inventories of release and waste data that are national in scope and issued on an annual, biannual, or triennial basis [1]. The Emissions & Generation Resource Integrated Database (eGRID) [2], the Toxics Release Inventory (TRI) [3], and the National Emissions Inventory (NEI) [4] are all examples of these inventories.

These inventories are each compiled independently to meet the requirements of different programs. For example, the Air Emissions Reporting Requirements (AERR) Rule in 40 CFR §51 requires states to report criteria air pollutants (CAPs) (e.g., carbon monoxide) every year for large "Type A" sources and every three years for "Type B" sources. If a

facility qualifies as a Type A or Type B source under the AERR, it will be incorporated in the NEI as a point source. The NEI contains specific reporting thresholds for Type A and Type B sources (details can be found in the supporting information). Reporting of hazardous air pollutants (HAPs) (e.g., acetaldehyde) is optional in the NEI. The reporting process in the NEI varies by state due to budget and by category of facility. The NEI is different from other inventories such as the TRI as the emissions are not necessarily reported directly by facilities. Emissions in the NEI can also be compiled by states, tribes, or the USEPA. The TRI covers facilities that manufacture, processes, or otherwise use any of the Emergency Planning and Community Right-to-Know Act (EPCRA) Section 313 chemicals. Facilities that have 10 or more full-time employees (as defined in 40 CFR §372.3) and are in a TRI-covered sector, as defined by the North American Industry Classification System (NAICS) code, or are a federal facility must report TRI chemical releases. Almost all HAPs reported within the NEI are also TRI chemicals. However, the TRI also includes toxic chemicals that may not be considered HAPs, and therefore are not assessed in the NEI. Unlike the NEI, TRI data are based on facility self-reporting, and facilities report on an annual basis. While there may be overlap in TRI and NEI air emissions, the TRI also reports chemical releases to other media such as land and water. In some cases, the NEI may use the TRI to supplement HAP information. However, the common air emission values between the NEI and the TRI may not always match due to differences in the reporting such as threshold values and the reporting process itself. States, tribes, and the USEPA have more discretion to modify NEI facility-level data during the reporting process as compared with the TRI reporting process. While each inventory adheres to relevant regulations, the differences in the coverage and reporting process can lead to complexities when using the databases together.

Public versions of these inventories are housed in various locations, are encoded in various data formats, and use various vocabularies to describe their contents. For example, eGRID data are released as Microsoft Excel workbook (xlsx) files for all facilities, while TRI data are hosted as a series of comma separated value (csv) files and are modified with updates regularly. Furthermore, several USEPA inventories are available through Envirofacts, a RESTful web service. As the facility-level inventory data files are generally very large (millions of records), accessing and working with them can require specialized knowledge. Over time, these inventories have also been changing to meet programmatic demands and are also stored and provisioned with new technologies.

These inventories provide critical information that helps to paint a national picture of environmental health and identify sources of potential environmental and human health issues related to pollution in the United States. They are widely used for modeling environmental conditions to assess environmental compliance (e.g., air quality modeling for meeting air quality standards) [5], determining needed capacity [6], developing benchmarks [7–10], evaluating time trends [11], and many other purposes [12,13]. Making these inventories easily usable may facilitate data compilation and modeling efforts.

The Standardized Emission and Waste Inventories (StEWI) tool is a set of Python packages written to support rapid and transparent processing of these inventories. More broadly, StEWI is one tool within an ecosystem developed by the USEPA to support modeling in the realm of industrial ecology [14]. StEWI performs consistent, reproducible processing and combination of USEPA emissions, releases, and waste inventories and adheres to principles used and developed in various modeling efforts such as the U.S. Environmentally Extended Input–Output (USEEIO) model [15] and the USEPA rapid life cycle inventory (LCI) [1]. While some of these inventories contain additional information, such as data aggregated by region, StEWI solely compiles the facility-based release and waste generation data from the inventories. Additionally, StEWI compiles and harmonizes metadata for the facilities and the pollutants or wastes.

StEWI is the first application of its kind known to the authors for rapid environmental data retrieval and combination from public sources for use in environmental modeling. Without StEWI, retrieving, processing, and understanding these inventories can be time consuming and prone to misinterpretation, particularly when applications require the

use of multiple inventories. In this regard, StEWI fills an important gap to enable further applications of these valuable data sources. This may be directly useful to reconcile and harmonize emissions from common U.S. inventory sources, but the approach that StEWI embodies may also be valuable for other regions in which reconciling multi-sector, multi-pollutant inventories is also a challenge [16].

Furthermore, StEWI adds value to individual inventory data in a number of ways. For example, StEWI can remove overlapping releases when more than one inventory reports the same release type. StEWI generates metadata such as flow reliability scores based on the quality of the data described in the original source using methods previously developed by Edelen and Ingwersen [17]. StEWI also provides a resource that other entities can easily use to generate facility-level environmental information. The modeling effort used here may be instructive for other integrated modeling efforts that compile large environmental datasets. StEWI is actively maintained as an open-source tool on GitHub and will be updated as new inventory years are released or data are revised.

The purpose of this article is to describe the structure of and data sources behind StEWI and to demonstrate the benefits of applying this novel package for analysis of a suite of facility-level emissions.

## 2. Materials and Methods

In this section, a general overview of StEWI's organization is given, followed by an in-depth explanation of the structure and function of each of the constituent libraries.

### 2.1. Organization and Dependencies

StEWI is designed as a semi-interdependent set of Python libraries (*stewi*, *chemical-matcher*, *facilitymatcher*, and *stewicombo*). Each library performs unique tasks in providing a standardized output or harmonization (Figure 1). Some libraries have dependencies on other libraries; all have a common structure and analogous Application Programming Interfaces (APIs). Like the other tools in the ecosystem, StEWI draws heavily on the *pandas* library [18] and uses the *pandas* dataframe as its basic structure for data storage, import, reshaping, and aggregation. *Pandas* is an extremely powerful data manipulation framework that has enabled the rapid rise of the data science field [19,20] The *requests* library is used generally to pull data from APIs when they are available [21]. Source data contained in Microsoft Excel files are read using the *openpyxl* engine (for .xlsx files) [22] or the *xlrd* engine (for .xls files) [23]. Outputs are stored in Apache parquet format [24], which enables efficient processing and retrieval of large datasets via the *pyarrow* library. Configurable elements for data retrieval and processing are generally stored in relevant .yaml files. YAML is a simple text-based format that is used to store configuration data across the tool ecosystem [14]. StEWI relies on the *PyYAML* package to parse these files [25]. Finally, StEWI relies on the USEPA LCA Ecosystem support package *esupy* for local file management, metadata processing, and path management.

### 2.2. Stewi

The *stewi* library consists of inventory-specific modules as well as common support modules that select, obtain, clean, and transform raw inventory data into standard output formats. (Note that StEWI is used to refer to the entire collection of libraries described herein, while *stewi* describes the individual library that directly accesses the raw inventory data). Each inventory module includes code used to process original sources into four standard outputs (flowbyfacility, flowbyprocess, facility, and flow) and record metadata. The formats are defined in the GitHub documentation under format specs, while the field names and data types associated with each format are defined in the formats.py module for use by *stewi*. Each standard file is processed for each inventory and for each year. URLs, file names, and other identifier information for retrieving inventory data sources are centrally stored as key–value pairs in config.yaml. Core functions in *stewi* for processing and retrieving processed data are highlighted in Table 1.

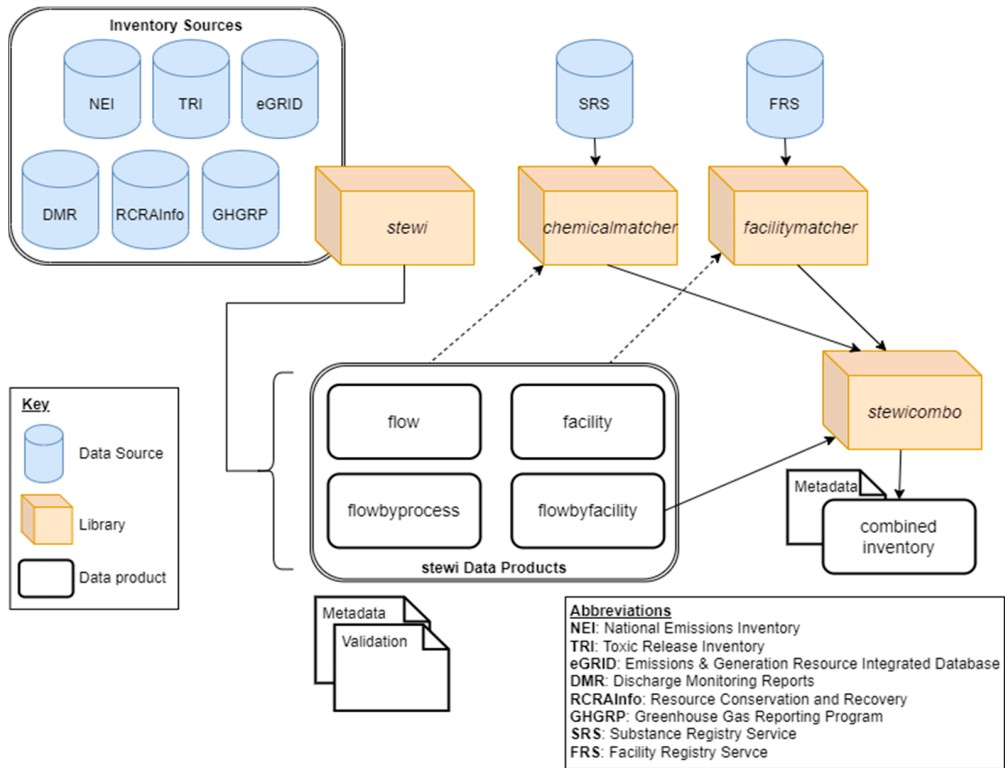

**Figure 1.** Standardized Emissions and Waste Inventory data flow.

**Table 1.** Core *stewi* functions for processing and accessing inventories.

| Function | Method |
|---|---|
| Get Available Inventories and Years | Returns a dictionary of processed inventory sources, where each key is a source and each dictionary value is a list of available processed data years. Uses flowbyfacility format by default. |
| Get Inventory | Returns a processed inventory as a dataframe in the standard output format. If that inventory does not exist locally, it will be generated. |
| Get Inventory Flows | Returns a processed flow inventory as a dataframe in the standard output format. If that inventory does not exist locally, it will be generated. |
| Get Inventory Facilities | Returns a processed facility inventory as a dataframe in the standard output format. If that inventory does not exist locally, it will be generated. |
| Get Metadata | Returns the metadata file from the local processed inventory. |

Data processed in *stewi* may include not just pollutant emissions but also inputs of resources and outputs of products or wastes. Therefore, the term 'flow' is adopted from the field of life cycle assessment [26] and is used to describe either a product, emission, or release of waste that is generated by an entity and enters into the environment or will be used or processed (e.g., waste treatment) by a downstream activity or entity. Each flow is assigned to a compartment, which reflects the media to which that flow is released (e.g., air, water, soil). The names and capitalization for flows given in the original inventory data are maintained. Additional metadata on the flows, such as the inventory ID for the flow and CAS number, if applicable, are stored in the unique set of flows for a given inventory and given year in the flow output file. All flow amounts are transformed from standard into

metric units, using kilograms (kg) for all mass flows and megaJoules (MJ) for energy flows. Conversation factors and conversion functions are stored in the globals.py module.

Data reliability scores are assigned to each flowbyfacility record using a method previously developed in Edelen et al. [17]. Meyer et al. [27] and Cashman et al. [1] describe the use of this method in the context of facility-level inventories. Upon processing, *stewi* assigns a data reliability score based on the method for deriving the flow value in the original inventory, with a flow reliability score of 1 representing a verified measurement, and a score of 5 representing the lowest data quality.

For each inventory processed, the flowbyfacility data totals are validated against reports of flow totals derived independently from the same inventory. The selected reports used as data sources for validation are called validation datasets. Depending on the reports available for each inventory, the flows are aggregated nationally or by state from the flowbyfacility outputs in order to compare these to the validation data. The totals by flows are compared against a calculated percent difference used as a tolerance level, where the default tolerance level is 5%. The result for each flow comparison is reported using the interpretations given in Table 2 and stored locally in a csv file. Comparisons where a data point is not found in either the processed inventory or the validation dataset are indicated as such. The code for the validation checks is contained in the validation.py module.

**Table 2.** Tolerance levels for validation. By default, the tolerance level (tl) is set to 0.05.

| Percent difference | Interpretation |
| --- | --- |
| 0.0 | identical |
| $\leq$tl | statistically similar |
| >tl | Percent difference exceeds tolerance |

*stewi* captures and records metadata on inventory sources, validation sources, and the output datasets. For inventory and validation source data, *stewi* records the filename, the URL the data were retrieved from, the date the data were retrieved, the file version, and the version of StEWI used to record the data. For output data, a standard class of source metadata defined in the *esupy* library is used, including fields for filename, output format (e.g., flowbyfacility), version of StEWI, git hash, and date created. Functions and defaults for metadata records are in the globals.py module.

Output files of type flowbyfacility and flowbyprocess can be filtered to remove records using a set of embedded filters when retrieved with getInventory. Filter names and instructions are stored in *filter.yaml* and implemented in functions in the filter.py module. Table 3 lists the filter, the inventories they apply to, and their functions. Embedded filters were created to generate StEWI output files for specific applications. Additionally, a filter_for_LCI parameter is available in getInventory that, when set to True, applies all the filters given in Table 3. All available filters can be printed to the console using see available inventory filters.

**Table 3.** Filters available for *stewi* outputs.

| Filter | Inventory | Function |
| --- | --- | --- |
| US_States_only | All | Removes data not assigned to facilities in one of the 50 U.S. States or D.C. |
| flows_for_LCI | TRI, DMR, NEI | Removes data for specific named flows that are not relevant for the LCI or would represent double counting |
| remove_duplicate_organic_enrichment | DMR | Removes overlapping organic enrichment reports. Facilities can report multiple forms of organic enrichment (BOD and COD), which represent duplicate reports of oxygen depletion (see Meyer et al. [27]) |

**Table 3.** *Cont.*

| Filter | Inventory | Function |
|---|---|---|
| National_Biennial_Report | RCRAInfo | Removes waste codes and facilities not associated with the National Biennial Report |
| imported_wastes | RCRAInfo | Removes imported wastes based on source code |

The following USEPA inventory sources are available for processing in *stewi*: the NEI (point source data only from the Emission Inventory System), the TRI, the eGRID, the Greenhouse Gas Reporting Program (GHGRP), Discharge Monitoring Reports (DMRs) based on reporting to the National Pollutant Discharge Elimination System, and the Resources Conservation and Recycling Act's Biennial Report generated from the RCRAInfo system (RCRAInfo). The processing of each of the inventory sources is described further below.

2.2.1. Discharge Monitoring Reports

Facilities report annual and sub-annual discharges to water under the Clean Water Act through the National Pollutant Discharge Elimination System (NPDES). The USEPA's DMR compiles data submitted by NPDES permit holders. The USEPA updates DMR flow quantities for facilities on an annual basis.

In the DMR.py module, *stewi* accesses DMR data via the Water Pollutant Loading Tool [28], a RESTful web service. Data for facilities are queried by state using the following query parameters:

- Flows are aggregated as "parameter groupings" to avoid double counting of flows that represent the same pollutant; this is especially relevant when facilities may be required to report multiple versions of the same release (e.g., different types of Chemical Oxygen Demand);
- The default setting for estimation is set to true; this setting estimates pollutant loads when no data are reported for a particular time period; and
- Non-detects are set to 50% of the detection limit.

Subsequently, aggregated nutrient quantities for nitrogen and phosphorous are queried by state with the Nutrient Aggregation feature on. With this feature, all nitrogen and phosphorus compounds are converted to N and P, respectively, equivalents based on a hierarchical evaluation in the Loading Tool to avoid double counting of reported nutrients [28].

Facility emissions are aggregated by state and validated against the State Statistics reported by the USEPA. The State Statistics only report emissions from NPDES Individual Permits and do not consider aggregated nutrients. So, the validation is performed prior to incorporating aggregated nutrients, and emissions captured by *stewi* from facilities with General Permits are excluded from the validation.

2.2.2. Emissions and Generation Resource Integrated Database

Through the eGRID, the USEPA compiles generation and emissions data for electricity-generating units in the United States [2]. These data are sourced from USEPA compiled statistics as well as facility-reported information to the Energy Information Administration (EIA). eGRID data are released semi-regularly, often every other year, in the form of Excel files. The specific emissions tracked by the eGRID are carbon dioxide ($CO_2$), nitrogen oxides ($NO_X$), sulfur dioxide ($SO_2$), methane ($CH_4$), and nitrous oxide ($N_2O$).

In the egrid.py module, *stewi* utilizes data from both the unit and plant-level datasets to parse the eGRID inventory. Plant level data tracked by *stewi* include annual emissions, heat input, and net generation. Where applicable, *stewi* also tracks the combined heat and power thermal output as steam. The unit level data supply the necessary information to characterize data reliability scores. Plant reliability scores for specific flows reflect the emission-weighted average of all units. While the eGRID reports generation mix by fuel

type, the emissions are reported as plant totals. As such, emissions are reported by facility in *stewi*, but the generation resource mix is maintained as additional facility metadata.

Facility emissions and generation are aggregated across all facilities and validated against national totals reported in the eGRID.

### 2.2.3. Greenhouse Gas Reporting Program

The GHGRP provides an inventory of greenhouse gases (GHG) at the facility and, in some cases, unit level [29]. Facilities with GHG emissions from covered sources that exceed 25,000 metric tons of $CO_2$ equivalent (eq.) per year must report to the GHGRP. Covered sources are listed by GHGRP subpart as documented in the Mandatory Greenhouse Gas Reporting rule in 40 CFR §98, Mandatory Greenhouse Gas Reporting. Example subparts include general stationary fuel combustion sources, electricity generation, ammonia production, aluminum manufacturing, ethanol production, petroleum refineries, and pulp and paper manufacturing. GHGRP reports for covered facilities are prepared on an annual basis.

Within the GHGRP.py module, *stewi* downloads a series of data tables containing GHGRP emissions data organized by subpart from the USEPA's Envirofacts API. Data from each subpart table are parsed to ensure a standardized format and concatenated into a master data table. With the GHGRP, emissions from stationary combustion sources (i.e., subpart C) can be estimated using one of four calculation methodologies, referred to as "tiers", plus one alternative methodology:

- The Tier 1 methodology uses default emission factors and high heating values to calculate mass emissions based on company records of fuel consumption;
- The Tier 2 methodology uses default emission factors to calculate mass emissions based on measured high heating values and company records of fuel consumption;
- The Tier 3 methodology calculates mass emissions based on measured fuel characteristics (e.g., carbon content, molecular weight) and measured fuel consumption;
- The Tier 4 methodology relies on a continuous emission monitoring system (CEMS) to calculate mass emissions from the stack gas concentrations and the stack gas flow rates; and
- In addition to these four methodologies, a small number of stationary combustion units may rely on 40 CFR §75 calculation methods based on monitoring data already collected under §75 (e.g., heat input, fuel use).

The emissions data from these five estimation methodologies are combined and organized into a standardized table categorized by gas. In some cases, data are reported at the unit level and must be aggregated to the facility level. Certain subparts (including subparts E, BB, CC, LL, L, and O) do not have their own standalone subpart tables and must be extracted from other data tables and parsed separately. After concatenating all subpart data into a master table, data are aggregated into standardized outputs that report GHG emissions by GHG flow (gas) and facility ID. Subpart data are maintained such that GHG data can be accessed in flowbyprocess format, which maintains total emissions by facility from each subpart. Data are validated against national-level data reported by the USEPA.

### 2.2.4. National Emissions Inventory

The NEI provides facility-level information on CAP and HAP emissions [4]. The AERR Rule in 40 CFR §51 requires States (via State, local, or tribal (S/L/T) entities) to report CAPs every year for large (Type A) point sources and every three years for other (Type B) point sources (Table S1). While facility reporting of HAPs is optional, the USEPA will augment facility-reported emissions with estimates based on speciation profiles or from the TRI. Table S2 provides the share of each method used for facilities reporting HAPs. Facilities report emissions data by source classification code (SCC), which corresponds to a standardized list of specific processes or emissions sources. NEI point sources may include large industrial facilities, electric power plants, and smaller industrial, non-industrial, and commercial facilities.



The NEI point source data are processed within the NEI.py module. *stewi* imports NEI data exported from the USEPA's Emissions Inventory System (EIS) Gateway. NEI data files are read into *stewi*, concatenated into a single data table, and parsed into a standardized format. Data reliability scores are assigned. Data are aggregated into standardized outputs that report emissions by flow and facility ID. Data in the NEI are also compiled in flowbyprocess format, which maintains reported emissions by facility for each unique SCC. Data are validated against national-level data reported by the USEPA.

### 2.2.5. Resource Conservation and Recovery Act Biennial Report

The Resource Conservation and Recovery Act Information (RCRAInfo) provides the type, disposition, and quantity of hazardous waste generated at the facility level. Facilities that treat, store, or dispose of hazardous waste must submit a Biennial Report [30] to RCRAInfo every two years.

Biennial Report data are downloaded by *stewi* from the USEPA's RCRAInfo Public Extract using the RCRAInfo.py module. Handler waste code descriptions are applied as flow names; where those waste codes are unavailable, form code descriptions are used instead. All facility and flow information is maintained in *stewi*, including wastes (e.g., imported wastes) and handlers not covered by the National Biennial Report. However, by default these handlers are filtered from the inventory upon accessing it via *stewi*. Data are validated against flow totals reported by State in the USEPA's Trends Analysis for the National Biennial Report.

### 2.2.6. Toxics Release Inventory

The TRI provides an inventory of air, water, and waste flows at the facility level for TRI-reportable chemicals only [3]. Facilities in the United States are required to report to the TRI if certain conditions are satisfied (e.g., they have 10 or more full-time employees, they are a TRI-covered sector as defined by the NAICS code, and the facility manufactures (defined to include importing), processes, or otherwise uses any EPCRA Section 313 chemical in quantities greater than the established threshold in the course of a calendar year). The TRI releases new inventory reports on an annual basis.

In the TRI.py module, *stewi* accesses TRI data through the Basic Plus data files, specifically files '1a: Facility, Chemical, Releases, and Other Waste Management Summary Information' and '3a: Details of Off-site Transfers'. Collectively, these files contain the facility and flow information necessary to characterize emissions and releases to air, water, and soil. While the TRI tracks transfers and the storage/management of covered chemicals, currently only exchanges directly with the environment are tracked in *stewi* (Table 4).

**Table 4.** TRI release types tracked by *stewi*.

| TRI Field | Source File | Compartment |
|---|---|---|
| ON-SITE—FUGITIVE AIR EMISSIONS | 1a | air |
| ON-SITE—STACK AIR EMISSIONS | 1a | air |
| ON-SITE—DISCHARGES TO STREAM | 1a | water |
| ON-SITE—LAND TREATMENT/APPLICATION FARMING | 1a | soil |
| ON-SITE—OTHER DISPOSAL | 1a | soil |
| OFF-SITE—LAND TREATMENT | 3a | soil |
| OFF-SITE—OTHER LAND DISPOSAL | 3a | soil |

Releases are aggregated across all facilities by flow and compartment and validated against national results from the TRI Explorer Release Chemical Report.

### 2.3. Chemical Matcher (Chemicalmatcher)

Each inventory reports a unique set of flows based on associated program requirements. The flows have unique nomenclatures, identifiers, and typographical conventions. The USEPA established the Substance Registry Service (SRS) for the purpose of centralizing

and cross-linking substances reported in USEPA inventories and other program systems. *chemicalmatcher* draws upon the flow output data from inventory processing from *stewi* and uses the SRS web services to gather a common identifier for the flows that can be used to link flows across the inventories. The *chemicalmatcher* output is similar to the *stewi* flow output, with the common identification number from the SRS (SRS_ID), formatted Chemical Abstract Services (CAS) number from the SRS (SRS_CAS), and inventory acronym (Source) added.

### 2.4. Facility Matcher (Facilitymatcher)

Like the flows reported by *chemicalmatcher*, *facilitymatcher* collects a unique set of data on facilities and identifies them with internal identifiers. *facilitymatcher* thus performs an analogous function to *chemicalmatcher* but for facilities, gathering a common facility identifier for cross-linking facilities across the inventories. The USEPA established the Facility Registry Service (FRS) for the purpose of providing common facility identifiers and facility information for facilities reporting to USEPA inventories and programs. The *facilitymatcher* output is in the form of the inventory facility identifier, (FacilityID), the common facility identifier (FRS_ID), and the inventory source (Source).

### 2.5. Combined Standardized Emission and Waste Inventories (stewicombo)

The *stewicombo* module utilizes the outputs of *stewi*, *chemicalmatcher*, and *facilitymatcher* to generate combined inventories that handle duplication and aggregation across inventory sources. In particular, *stewicombo*:

- Identifies common facilities across datasets;
- Aggregates multiple entities into a single facility;
- Assesses potential duplicate flows from a facility when a flow is reported to be emitted to the same compartment across more than one inventory; and
- Enables custom handling of inventories (e.g., inventory preferences for duplicate flows).

In *stewicombo*, users have access to three methods for combining inventories (Table 5). These methods allow users to combine flowbyfacility data from one or more inventories. In each case, *stewicombo* first aligns facilities across inventories using the FRS_ID sourced from *facilitymatcher* and then aligns flows using the SRS_ID from *chemicalmatcher*.

**Table 5.** Functions for combining inventories in *stewicombo*.

| Function | Method |
|---|---|
| Combine Full Inventories | Combines flowbyfacility data for all facilities in the selected inventory(-ies) |
| Combine Inventories for Facilities in Base Inventory | Combines flowbyfacility data for all facilities in the selected inventory(-ies); maintains only those facilities with data present in the base_inventory. |
| Combine Inventories for Facility List | Combines flowbyfacility data for all facilities in the selected inventory(-ies); maintains only those facilities with data present in the base_inventory that are included in the facility_id_list. |

The overlap handler module of *stewicombo* handles the core functions of aggregating and removing overlapping flows within facilities. Under default settings, inventory records are compiled using the following logic:

- Records that share a common compartment and SRS_ID (i.e., are the same flow) and FRS_ID (i.e., are the same facility) within an inventory are summed. This case typically reflects a single facility reporting to two or more facility IDs within an inventory that need to be aggregated; and
- Records that share a common compartment and SRS_ID (i.e., are the same flow) and FRS_ID (i.e., are the same facility) across multiple inventories are assessed by

compartment preference (see Table 6). This case reflects double counting by reporting of the same chemical across two or more inventories.

**Table 6.** Default inventory preference by compartment, as documented in the parameter INVENTORY_PREFERENCE_BY_COMPARTMENT.

| Compartment | Inventory Preference |
|---|---|
| air | (1) eGRID, (2) GHGRP, (3) NEI, (4) TRI |
| water | (1) DMR, (2) TRI |
| soil | (1) TRI |
| waste | (1) RCRAInfo, (2) TRI * |

* Chemical quantities of waste from the TRI are not yet handled by *stewi*. Note that RCRAInfo reports waste quantities and not chemical quantities.

Additional steps are taken to avoid overlap of:

- Nutrient flow releases to water between the TRI and DMR;
- Particulate matter releases to air reflecting PM < 10 and PM < 2.5 in the NEI; and
- Volatile Organic Compound (VOC) releases to air for individually reported VOCs and grouped VOCs.

## 3. Results

StEWI v1.0 was used to produce the flowbyfacility, flow, and facility files for the inventories presented in Table 7.

**Table 7.** USEPA inventories accessible by StEWI.

| Source | 2008 | 2009 | 2010 | 2011 | 2012 | 2013 | 2014 | 2015 | 2016 | 2017 | 2018 | 2019 |
|---|---|---|---|---|---|---|---|---|---|---|---|---|
| Discharge Monitoring Reports * | | | | | | | x | x | x | x | x | x |
| Greenhouse Gas Reporting Program | | | | x | x | x | x | x | x | x | x | x |
| Emissions & Generation Resource Integrated Database | | | | | | | x | | x | | x | x |
| National Emissions Inventory ** | | | | x | *i* | *i* | x | *i* | *i* | x | *i* | |
| RCRA Biennial Report * | | x | | x | | x | | x | | x | | x |
| Toxics Release Inventory * | x | x | x | x | x | x | x | x | x | x | x | x |

* Earlier data exist and are accessible but have not been validated. ** Only point sources included at this time from the NEI; *i* interim years between triennial releases, accessed through the Emissions Inventory System, have not been validated.

StEWI provides a meaningful way to track facility-based emission, energy, and waste flows over time and across inventories. Assessment of flows across inventories allows for a more thorough understanding of trends that are agnostic to any particular inventory. As an example, changes in selected flows from selected inventories since 2014 are shown in Figure 2.

Visualizing the same flow from multiple inventories demonstrates the benefit of using *stewicombo* to combine inventory data for the same or related flows. Ammonia, nitrogen dioxide, nitrogen oxides, and nitrogen are all nitrogen-derived pollutants that can have negative impacts on human and ecosystem health. While some of the air and ground-based species show relative stability with annual fluctuations or a slight decline over 2014–2018, the nitrogen released to water as reported in the DMR shows a large relative increase.

Similarly, three inventories report releases of carbon dioxide ($CO_2$) from facilities (Figure 3). The eGRID only includes emissions from electricity-generating units, which are also covered in the GHGRP. Since 2016, facility emissions in the GHGRP have also been included in data made available through the NEI. The use of *stewicombo* provides increased certainty that these emissions are not double-counted across the inventories, while also providing a more consistent time series, since data are not available in all inventories each year. Under the default settings for *stewicombo*, emissions from facilities are sourced first from the eGRID (i.e., for electricity-generating units), and then from the GHGRP, prior to

including any remaining GHG emissions from the NEI not already included. With the exception of the first year of inclusion for the NEI, all three datasets show a very similar trend of decreasing facility emissions across the time period.

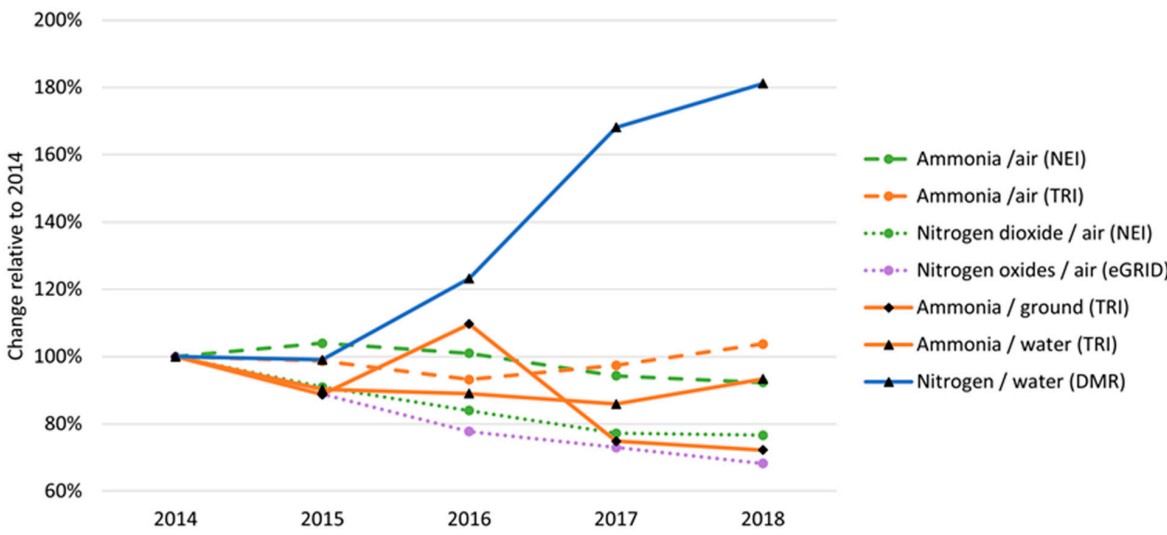

**Figure 2.** Tracking national emissions by inventory over time.

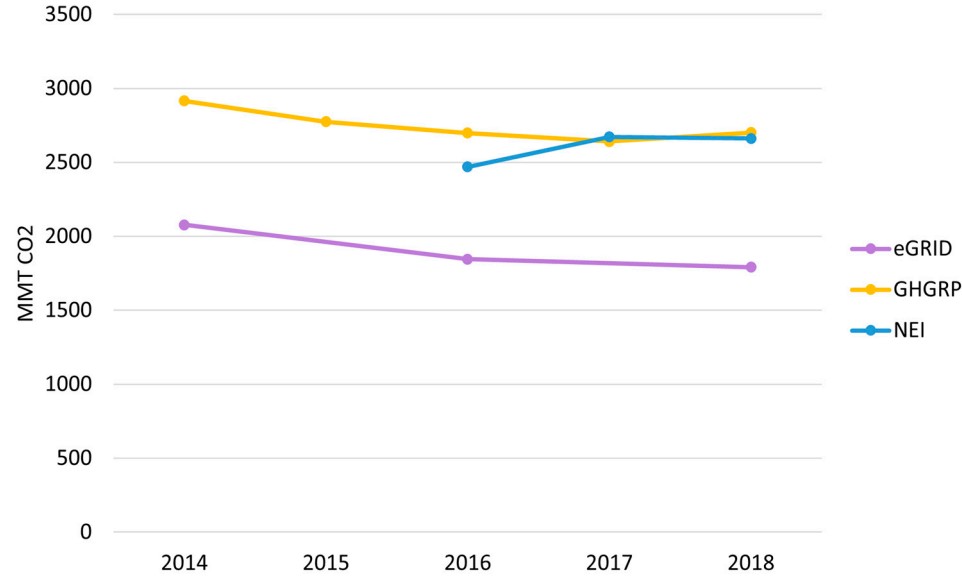

**Figure 3.** National carbon dioxide emissions by inventory over time.

Figure 4 highlights the reduction in total flows using *stewicombo* in comparison with the data reflecting the original reformatted and harmonized totals from the inventories, represented in the StEWI bar. The equal or lower total amounts reflected in the *stewicombo* results show the effect of the removal of overlapping flows. For example, based on the default preference of NEI data over TRI data for emissions to air, the vast majority of TRI-sourced ammonia emissions to air are removed from the combined inventory (the first pair of bars). However, because the TRI is the sole source of emissions data for ammonia to ground, no data are removed from that pool of emissions through the use of *stewicombo*. *chemicalmatcher* identifies potential overlaps in flows between nitrogen dioxide (NEI) and ammonia (TRI) and thus some flows are removed based on inventory preferences.

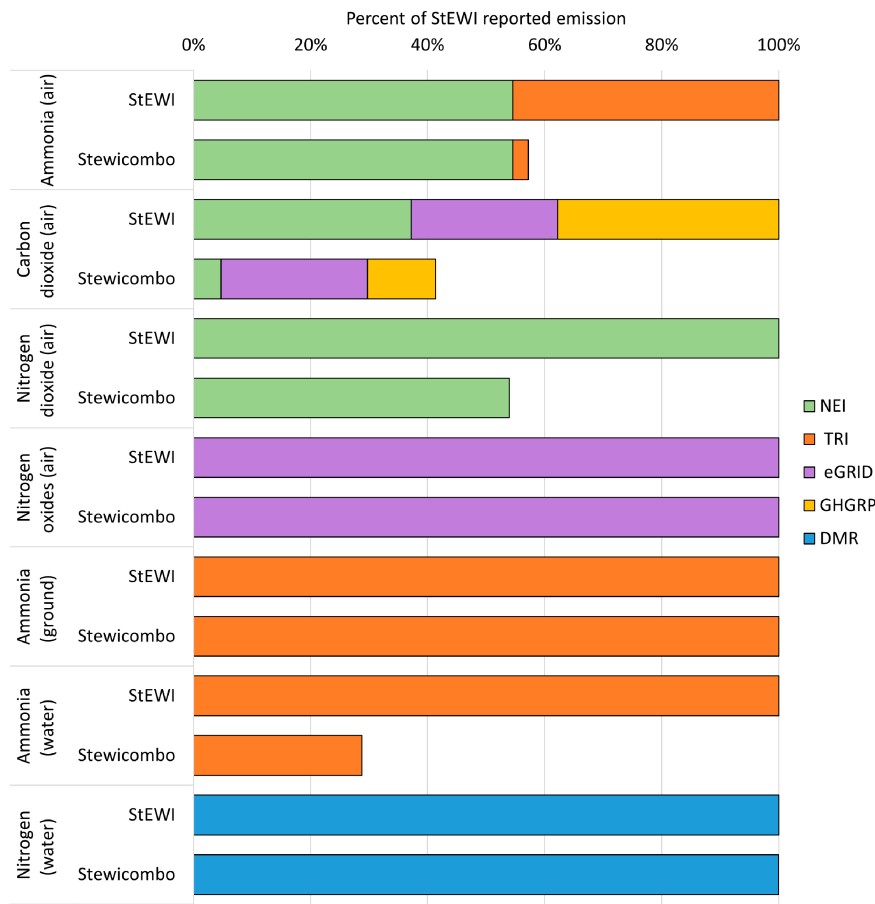

**Figure 4.** Reducing overlap with *stewicombo* (reporting year 2018).

By utilizing other tools in the USEPA LCA tool ecosystem, the potential environmental and human health impacts from reported releases can be assessed. Flows from each inventory are mapped to the Federal Elementary Flow List [31] and potential environmental and human health impacts are assessed by pairing the flows with characterization factors from the USEPA's Tool for Reduction and Assessment of Chemicals and Other Environmental Impacts (TRACI) v2.1 from the LCIA Formatter [32]. All the releases to water, air, and ground from each inventory, including the eGRID, GHGRP, NEI, DMR, and TRI, are multiplied by corresponding characterization factors from each applicable impact category and then the impacts from each flow are summed together within each impact category to calculate a total impact for each inventory in the respective category. Figure 5 shows the relative contribution for a subset of flows to impacts from all inventories in 2018. The bar on the left shows the impact distribution prior to using the overlap handler in *stewicombo*. For example, total global warming potential would be significantly over-estimated when combining data from the NEI, GHGRP, and eGRID without the use of *stewicombo*. Likewise, other impact categories show a significant amount of overlap across inventories, including acidification potential and smog formation potential.

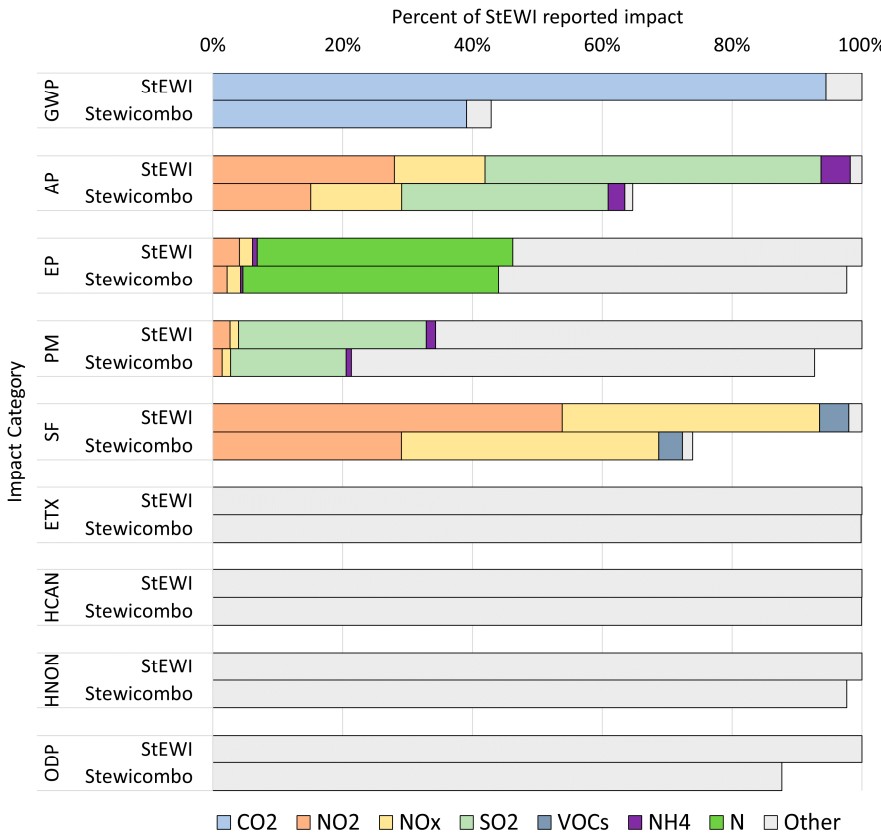

**Figure 5.** National contribution of Life Cycle Impacts by Flow in 2018. GWP, global warming potential; AP, acidification potential; EP, eutrophication potential; PM, particulate matter formation; SF, smog formation; ETX, ecotoxicity; HCAN, human-health cancer; HNON, human-health noncancer; ODP, ozone depletion potential.

## 4. Discussion

The StEWI package and data products are already being used for studies modeling chemical releases [33] and tracking hazardous industrial wastes [34]. StEWI is also being used to develop "rapid" or automated life-cycle inventory models and create environmental accounts for environmentally extended input–output models. Point source releases of flows compiled by StEWI are used to supplement emissions data for the U.S. Electricity LCI Baseline [35] by providing emissions to air, water, and soil, as well as hazardous waste flows for all U.S. electricity-generating units. EIA data for electricity-generating units are readily available but typically only track a small subset of environmental flows preferred for life cycle modeling. Data from the NEI, TRI, and RCRAInfo provide fuller coverage for a wider range of flows, while data from the eGRID provide an alternate data source for validation. The USEEIO model relies on StEWI for point source emissions data to better characterize environmental impacts of U.S. industries [36]. The python package FLOWSA compiles facility emissions data from StEWI and aggregates them based on the NAICS code assigned for each facility in *facilitymatcher* [37]. The resulting industry emissions totals are combined with emissions data from other sources to obtain broad environmental coverage for the USEEIO model. In both cases, *stewicombo* helps prevent double counting of emissions flows for data reported across inventories.

With several active applications, StEWI is expected to be further expanded to support more advanced emissions tracking and life cycle modeling. For example, data provided in some inventories will enable more refined processing of emission compartments. Stack height for air emissions is relevant for air quality modeling and life cycle impact assessment. Additionally, most inventories provide geographic coordinates for facilities, which can be compared to maps of population density and better characterize the human health

impacts of releases. Furthermore, while StEWI currently only supports U.S.-based inventory sources from the USEPA, the framework and data structures could also be adapted to inventories from other countries. These updates, as well as other expansions to emissions compartments, are expected in future releases of StEWI.

## 5. Conclusions

StEWI provides a much-needed resource to enable users to access, process, and apply USEPA inventory datasets for a wide variety of applications. The Python packages that make up StEWI support transparent and replicable data processing without demanding expert inventory knowledge from users. As an open-source software package available on GitHub, and a key resource within the USEPA LCA tool ecosystem, StEWI is actively maintained and designed to support expanded features for the broader research community.

**Supplementary Materials:** The following supporting information can be downloaded at: https://www.mdpi.com/article/10.3390/app12073447/s1, Table S1: Emissions Thresholds in Tons per Year for Type A Sources, Type B Sources, and Sources in Nonattainment areas, Table S2: Percent of facilities that report Hazardous Air Pollutants by method. S/L/T: State, local, or tribal agency.

**Author Contributions:** Conceptualization, W.W.I.; software, B.Y., W.W.I., M.B., J.D.H.-B., T.G. and E.B.; data curation, B.Y., W.W.I., M.B., J.D.H.-B. and T.G.; investigation, B.Y., W.W.I., M.B., J.D.H.-B. and T.G.; methodology, S.C., W.W.I., J.D.H.-B. and B.Y.; supervision, W.W.I. and S.C.; validation, B.Y. and W.W.I.; visualization, B.Y.; writing, B.Y., W.W.I., S.C. and E.B. All authors have read and agreed to the published version of the manuscript.

**Funding:** Funding for StEWI came from the USEPA Office of Research and Development's Chemical Safety and Sustainability national research program and from the former National Risk Management Research Laboratory. This research was supported through USEPA contract EP-C-16-015, task order 68HERC19F0292 with ERG, contract 68HERC21D0003, task order 68HERC21F0133 with ERG, and contract EP-C-15-012 and work assignment 20 with CSRA. The Research Participation Program, administered by the Oak Ridge Institute for Science and Education through an Interagency Agreement between the U.S. Department of Energy and the USEPA, supported the appointment of Jose D. Hernandez-Betancur.

**Data Availability Statement:** Data generated in this study are available at https://edap-ord-data-commons.s3.amazonaws.com/index.html?prefix=stewi/ (accessed on 22 February 2022). Source code for StEWI is available at https://github.com/USEPA/standardizedinventories (accessed on 22 February 2022).

**Acknowledgments:** David Meyer's work on rapid life inventory modeling using EPA inventories (EPA) paved the way for the automation of inventory processing embodied in StEWI and he helped coin the name 'StEWI'. Stacie Enoch (ERG), Eva Knoth (ERG), Jennifer Sellers (ERG), Stacy DeGabriele (ERG), and Cpan Lee (USEPA) provided expertise in the use of EPA inventories. Andy Chase (USEPA), David Graham (USEPA), William Barrett (USEPA), and Hui Zou (ERG) reviewed software code. Jorge Rangel (USEPA), Nancy Parrota (USEPA), and Bill Michaud (GDIT) assisted with contract management. Darcie Smith (USEPA), Kristin Isaacs (USEPA), Gerardo Ruiz Mercado (USEPA), and Michael Gonzalez (USEPA) assisted with research management. Catherine Birney (USEPA), Mo Li (GDIT), Andrew Beck (ERG), Mike Liadov (ERG), Alexandra Starr (GDIT), and Greg Schively provided additional coding support and insight.

**Conflicts of Interest:** The authors declare no conflict of interest.

**Disclaimer:** The U.S. Environmental Protection Agency, through its Office of Research and Development, funded and conducted the research described herein under an approved Quality Assurance Project Plan (K-LRTD-0033145-QP-1-1). It has been subjected to review by the Office of Research and Development and approved for publication. Approval does not signify that the contents reflect the views of the Agency, nor does mention of trade names or commercial products constitute endorsement or recommendation for use.

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
