# Peer review of "A System for Standardizing and Combining U.S. Environmental Protection Agency Emissions and Waste Inventory Data"

_applsci, doi:10.3390/app12073447_

Round 1

Reviewer 1 Report

No specific comments.

Author Response

Response to Reviewer 1 Comments

No specific comments

Thank your for your review.

Reviewer 2 Report

This paper needed to mention the following issues before going forward:

  • Abstract:
    • Add some numerical results.
  • Introduction:
    • The introduction should be included more references.
    • A paragraph should be presented to show the importance of this study. What is new in this study? The authors should be stated the importance of their work.    
  • The author should add the “conclusion” part to their manuscript to show the limitations of their study.    

Author Response

Response to Reviewer 2 Comments

Point 1. Abstract: Add some numerical results.

The abstract was revised including to add numerical examples of the benefits of StEWI for emissions totals.

Point 2. The introduction should be included more references.

Additional references were added to the introduction especially to highlight other uses of these databases in the literature for a variety of applications.

Point 3. A paragraph should be presented to show the importance of this study. What is new in this study?

Text was added to the introduction to discuss explicitly the importance and newness of the StEWI package and the gaps it fills in the broader research community.

Point 4. The authors should be stated the importance of their work.

See point 3 above.

Point 5. The author should add the “conclusion” part to their manuscript to show the limitations of their study.

A conclusion section was added to summarize the contributions of StEWI to the broader community.

Author Response

Response to Reviewer 3 Comments

Point 1. The text is written in an orderly and clear manner. The individual elements that make up the StEWI package have been described and explained in detail. The authors clearly presented the context related to the need to create such a tool in the introduction. What I lacked was only the diagrams that illustrate the relationship between the described elements in a graphical form, which concerns both the data sources and the structure of the StEWI system itself.

Thank you for this suggestion, we have added a summary graphic to the beginning of the methods to highlight the data structure and data sources.

Point 2. There is no summary chapter in the text. Its fragments appear in the discussion, but it will be difficult for the reader to point directly to the conclusions of the presented work.

A conclusion section was added to summarize the contributions of StEWI to the broader community.

Point 3. References need review. The list looks a bit like a draft. I did not find this article:

Ingwersen, Wesley, Catherine Birney, Ben Young, Mo Li, and Sarah Cashman. 2022. “An Open-Source Tool Ecosystem for Industrial Ecology.” Applied Sciences.

References to unpublished materials were removed and replaced with alternate citations. The reference style was updated to match Applied Sciences format.

Point 4. The number of keywords is way too high.

We removed the acronyms for the data sources. There a large number because the software describes integrates a lot of data sources, and we would like potential researchers to find this paper if they are looking for information about data from those sources. We further defer to the editor to determine if there remain too many and if given a limit, we will remove more. Otherwise we prefer to leave the remaining keywords.

Point 5. It is a pity that the article does not explicitly provide the address of the project’s repository (only a hyperlink) and the license.

The github link to the project was made explicit in the introduction.

Point 6. What raises my doubts is starting a lot of sentences with the word “stewi”. This is not the package name (character size). Sentences do not start with a lowercase letter. It generally looks and reads badly.

stewi (lowercase, italics) is the name of one of the libraries within the broader Standardized Emissions and Waste Inventories tool. For better readibility, we have modified sentence structure to avoid starting sentences with the library name.

Point 7. Typo in Figure 4 caption (repoted).

Typo fixed.

Reviewer 4 Report

The manuscript “A System for Standardizing and Combining US Environmental Protection Agency Emissions and Waste Inventory Data” sent to Applied Sciences (applsci-1631144) presents a study about the system called Standardized Emission and Waste Inventories (StEWI), which contain four python modules that provide rapid access to The U.S. Environmental Protection Agency inventory data in standard formats and permit filtering and combination of these inventory data.

The subject is in the scope of Applied Sciences.  In general, the article is written understandably. The problem and research methods are clearly specified. It is devoted to a current, interesting issue.

This manuscript can be accepted for publication after the following minor corrections:

  1. The abstract is longer than 200 words and includes introductory general information.
  2. Is the presented research can be applied outside the US?
  3. stewi or StEWI (for example line 131)?
  4. The Conclusion chapter should be created. The Discussion section should be rewritten because it contains information that summarizes the article.
  5. References on the list should be numbered.
  6. There is no link to GitHub website.

Author Response

Response to Reviewer 4 Comments

Point 1. The abstract is longer than 200 words and includes introductory general information.

The abstract was shortened and introductory information was reduced.

Point 2. Is the presented research can be applied outside the US?

A sentence was added in the discussion clarifying that only U.S. inventory sources are currently included but the framework could be applied outside the U.S.

Point 3. stewi or StEWI (for example line 131)?

StEWI is used to refer to the entire collection of libraries described herein, while stewi describes the individual library that directly accesses the raw inventory data. This clarification was added to the text and a few fixes to this nomenclature were made.

Point 4. The Conclusion chapter should be created. The Discussion section should be rewritten because it contains information that summarizes the article.

A conclusion section was added to summarize the contributions of StEWI to the broader community.

Point 5. References on the list should be numbered.

Reference list was updated to match the Applied Sciences format using reference numbers.

Point 6. There is no link to GitHub website.

The github link to the project was made explicit in the introduction.

Round 2

Reviewer 2 Report

The authors address well all reviewer comments.